# Modeling critical thermoelectric transports driven by band broadening and phonon softening

Kunpeng Zhao [1,2,5] ✉, Zhongmou Yue[1,3,5], Hexige Wuliji[2], Hongyi Chen [4] ✉, Tingting Deng[1,3], Jingdan Lei [2], Pengfei Qiu [1,3], Lidong Chen [1,3] & Xun Shi [1,3] ✉

Critical phenomena are one of the most captivating areas of modern physics, whereas the relevant experimental and theoretical studies are still very challenging. Particularly, the underlying mechanism behind the anomalous thermoelectric properties during critical phase transitions remains elusive, i.e., the current theoretical models for critical electrical transports are either qualitative or solely focused on a specific transport parameter. Herein, we develop a quantitative theory to model the electrical transports during critical phase transitions by incorporating both the band broadening effect and carrier-soft TO phonon interactions. It is found that the band-broadening effect contributes an additional term to Seebeck coefficient, while the carrier−soft TO phonon interactions greatly affects both electrical resistivity and Seebeck coefficient. The universality and validity of our model are well confirmed by experimental data. Furthermore, the features of critical phase transitions are effectively tuned. For example, alloying S in $Cu_2Se$ can reduce the phase transition temperature but increase the phase transition parameter $b$. The maximum thermoelectric figure of merit $zT$ is pushed to a high value of 1.3 at the critical point (377 K), which is at least twice as large as those of normal static phases. This work not only provides a clear picture of the critical electrical transports but also presents new guidelines for future studies in this exciting area.

Critical phenomena are common in nature[1,2], such as the increase in compressibility upon approaching the critical point of liquid-vapor equilibrium[3], the increase in magnetic susceptibility and dielectric constant in the vicinity of the Curie points of ferromagnets and feorroelectrics[4], the anomaly in heat capacity at the point of critical transition[5], and the abnormally thermoelectric (TE) properties at the point of critical phase transition[6–13]. Particularly, the materials

undergoing critical phase transitions have pronounced atomic disorder and critical fluctuations (Fig. 1a), which bring about a rich diversity of electrical and thermal transports near the critical point[6–19]. For instance, anomalous reduction in thermal conductivity $\kappa$ has been observed during critical phase transitions for a lot of materials, such as $Cu_2Se$[6,15] and $BaTiO_3$[20]. Dramatic increment in $\rho$ have been reported around the ferroelectric phase transitions of SnTe[16,21], PbTe[22], GeTe[8],

[1]State Key Laboratory of High Performance Ceramics and Superfine Microstructure, Shanghai Institute of Ceramics, Chinese Academy of Sciences, Shanghai 200050, China. [2]State Key Laboratory of Metal Matrix Composites, School of Materials Science and Engineering, Shanghai Jiao Tong University, Shanghai 200240, China. [3]Center of Materials Science and Optoelectronics Engineering, University of Chinese Academy of Sciences, 100049 Beijing, China. [4]College of Chemistry and Chemical Engineering, Central South University, Changsha, Hunan 410083, China. [5]These authors contributed equally: Kunpeng Zhao, Zhongmou Yue. ✉e-mail: zkp.1989@sjtu.edu.cn; hongyichen@csu.edu.cn; xshi@mail.sic.ac.cn

**Fig. 1 | Band broadening and phonon softening by structural fluctuation during critical second-order phase transitions. a** Schematic map of structural fluctuation, (**b**) Free energy $\Phi$ as a function of order parameter $\xi$ for different $b$ or temperatures. Schematic maps of (**c**) band broadening and (**d**) phonon softening effects. **e** Illustration of the relatively enhanced Seebeck coefficient ($\Delta\alpha/\alpha_0$) driven by band broadening and phonon softening effects (red indicating large $\Delta\alpha/\alpha_0$ and purple indicating small $\Delta\alpha/\alpha_0$, as shown in the scale on the right), with respect to the temperature ($T$) and phase transition parameter $b$. **f** The relatively enhanced electrical resistivity ($\Delta\rho/\rho_0$) attributed by carrier-soft TO phonon interaction, with respect to the temperature ($T$) and phase transition parameter $b$.

and their ternary alloys. Likewise, sharp enlargements in both $\rho$ and $\alpha$ were observed in the vicinity of phase transition of $Cu_2Se$[6,15,18,23–25]. Combining these effects, abnormally enhanced TE figure of merit ($zT = \alpha^2 T/\rho\kappa$) with the $\lambda$ shape during critical phase transitions is obtained, alike the other critical physical properties during phase transitions. The mechanisms for the critical TE transports have been also studied recently, such as the strong critical scattering[6,15], soft transverse-optical (TO) modes[26–30], the increased structure entropy[17,18], broad energy gap[31], and gradually changed electronic states[32]. However, in most cases, these models are typically qualitative or just consider specific mechanisms. Furthermore, many of them only solely focus on either the electrical resistivity or the Seebeck coefficient,

rather than the whole electrical transports. A quantitative and comprehensive model covering both electrical conductivity and Seebeck coefficient during critical phase transitions is still absent until today, which greatly restricts the understanding of the physics involved and hampers the discovery of new critical TE materials.

In this work, starting from the Landau theory, we developed a theoretical model to quantitatively describe both electrical resistivity and Seebeck coefficient during critical phase transitions by incorporating the band broadening effect and carrier-soft TO phonon interactions. Subsequently, this model is successfully validated by experimental data and serves as a useful guide for designing novel TE materials with critical transports.

## Results and discussion

### Modeling of the critical electrical transports

A second-order phase transition is generally critical, which refers to the continuous structural transition from a low-temperature ordered phase to a high-temperature disordered phase accompanied by continuous change of entropy and volume[33–35]. The order parameter $\xi$, a measure of the character and strength of the broken symmetry, was proposed by Landau to describe the order-disorder transition (see details in the Supplementary Information)[35]. Generally, $\xi$ has a zero value above the transition temperature and a nonzero value below the transition temperature. When temperature approaches the critical phase transition point $T_p$, the $\xi$ fluctuates in a range of $\xi_0(T) \pm \Delta\xi$, where the distribution of $\Delta\xi$ follows the Maxwell-Boltzmann distribution[35]

$$w(\Delta\xi) = A_\xi e^{-\frac{\Delta\Phi}{k_B T}} \tag{1}$$

Here $A_\xi$ is a normalized constant, $k_B$ is the Boltzmann constant, $T$ is the absolute temperature, and $\Delta\Phi$ is the energy required for the fluctuation of $\xi$, which is approximate to $\Delta\Phi \sim 2b \times (T-T_p) \times (\Delta\xi)^2$ for a small departure from the equilibrium state. As shown in Fig. 1b, fewer energy is required for the fluctuation of $\xi$ when phase transition parameter $b$ is small or temperature is close to $T_p$.

The variation of $\xi$ implies the critical fluctuations in atoms site, lattice strain, and crystal symmetry (Fig. 1a), which would lead to two important effects. One is the broadening or smearing of band-edge density of states[7,36] (Fig. 1c), and the other is the softening of transverse-optic (TO) phonons[37,38] (Fig. 1d). Here we try to quantitatively give the formula how these effects influence the electrical resistivity and Seebeck coefficient. Based on the classical Landau theory[33,35], the distribution function of the band energy $E$ is given by

$$w(E) = A_\xi e^{-\frac{(E-E_0)^2}{\Delta^2}} \tag{2}$$

where $\Delta$ is a broadening function and given by

$$\Delta = \left(\frac{k_B c^2 T}{2b(T_p - T)}\right)^{\frac{1}{2}} \tag{3}$$

Here $c$ is the first derivate of energy with respect to order parameter. The derived Gaussian distribution of band broadening and the temperature dependency of broadening function $\Delta$ are well consistent with the band-smearing effect in RbAg4I5 reported by Bauer et al.[36] A small $b$ value or a temperature close to $T_p$ obviously lead to a large broadening of band-edge energy. Mahan[31] presented a theory for the anomalous critical Seebeck coefficient based on a fitted $\Delta$ from experimental data. After substitution of $\Delta$ (i.e. Eq. (3)) into Mahan's derivation, the formulas for the material with band broadening is obtained

$$\frac{\Delta\alpha_1}{\alpha_0} = \frac{3c^2}{8\pi^2 b} \frac{1}{T T_p (1 - T/T_p)^2} \tag{4}$$

$$\rho = \frac{3}{2e^2} \frac{1}{\Sigma(\eta)} \tag{5}$$

where $\alpha_0$ is the normal Seebeck coefficient of a conductor[39], and $\Delta\alpha_1$ is the enhanced Seebeck coefficient from band broadening. $\Sigma(\eta) = \int \frac{d^3k}{(2\pi)^3} v_k^2 \tau(k) \delta(k)$, where $v_k$ is the electronic wave factor, and $\tau_k$ is the relaxation time of electrons. Clearly, the band broadening effect during critical phase transitions has no impact on electrical resistivity, but contributes an additional term to Seebeck coefficient. The enhanced $\Delta\alpha_1$ strongly depends on the temperature $T$ and phase

transition parameter $b$. At the critical point $T_p$, the maximum ratio $\frac{\Delta\alpha_{1,max}}{\alpha_0}$ approaches infinity, which, however, is unable to be determined in experiments because the Seebeck coefficient is measured within a certain temperature difference range. Below $T_p$, $\frac{\Delta\alpha_1}{\alpha_0}$ is inversely proportional to $b$, i.e. a smaller $b$ would yield a larger Seebeck coefficient.

Now we turn to study the effect of softening of transverse-optic (TO) phonons on critical electrical transports. Based on the standard theory of the second-order phase transition, the soft TO phonon frequency $\omega$ decreases rapidly when $T$ is approaching to $T_p$ (Fig. 1d)[40]

$$\omega^2 = b(T_p - T) \tag{6}$$

The enhanced electrical resistivity $\Delta\rho_2$ caused by phonon softening (see the derivation details in the Supplementary Information) is then given by

$$\frac{\Delta\rho_2}{\rho_0} = C_1 T \left[1 - \frac{b(T_C - T)}{\gamma} \ln\left(1 + \frac{\gamma}{b(T_C - T)}\right)\right] \tag{7}$$

Here $C_1 = \frac{\sqrt{2}}{16} \frac{k_B}{MD} \left|\frac{V_p}{a}\right|^2 \frac{m^{\frac{1}{2}} \tau_0 E_F^{-\frac{1}{2}+s}}{\hbar^2}$, which is a parameter independent on temperature $T$ and phase transition parameter $b$, $V_p$ is the potential energy of electron-phonon interaction, and $\gamma$ is related to the dispersion coefficient $D$ and Fermi vector of electrons $k_F$, which is given by $\gamma = 4Dk_F$[28]. $s$ represents the scattering factor and takes the value of −0.5 for acoustic phonon scattering and 0.5 for optical phonon scattering. When $T$ is approaching to $T_p$ or $b$ is reduced, $\Delta\rho_2$ is significantly enhanced due to the strong interaction between carriers and soft TO phonons, as illustrated in Fig. 1f. At the critical point $T_p$, the maximum ratio $(\frac{\Delta\rho_{max}}{\rho_0} = C_1 T_p)$ is obtained, which is independent on parameter $b$. This is understandable because the TO phonon frequency $\omega$ becomes zero at the critical point irrespective of $b$ (see Eq. 6).

Furtherly, the impact of phonon softening on Seebeck coefficient can be determined by solving the Mott expression (see the derivation details in the Supplementary Information):

$$\frac{\Delta\alpha_2}{\alpha_0} = \frac{\Delta\rho_2/\rho_0}{1 + \Delta\rho_2/\rho_0} \frac{\frac{1}{2} - s}{s + \frac{3}{2}} \tag{8}$$

where $\Delta\alpha_2$ is the additional Seebeck coefficient contributed by the phonon softening. It is clear that $\Delta\alpha_2$ arises from the change of scattering mechanism. If the electrical conduction of the normal phase is dominated by the optical phonon scattering (i.e. $s = 0.5$), $\Delta\alpha_2$ becomes zero due to the fact that phase transition-induced electrical transport is also governed by optical phonon scattering.

Incorporating the band broadening and phonon softening effects, the enhanced Seebeck coefficient $\Delta\alpha$ during critical phase transition is given by:

$$\frac{\Delta\alpha}{\alpha_0} = \frac{3c^2}{8\pi^2 b} \frac{1}{T T_p (1 - T/T_p)^2} + \frac{\Delta\rho_2/\rho_0}{1 + \Delta\rho_2/\rho_0} \frac{\frac{1}{2} - s}{\frac{3}{2} + s} \tag{9}$$

Equations (7) and (9) are the quantitively relationship for critical electrical transports ($\alpha$ and $\rho$), which are strongly correlated via parameter $b$. As the temperature approaches $T_p$ or $b$ is reduced, the fluctuation of order parameter $\xi$ becomes critical, leading to much increased Seebeck coefficient and electrical resistivity, as illustrated in Figs. 1e and 1f.

It should be noted that there are certain approximations or biases in the modeling. Firstly, we only considered the second power of the order parameter $\xi$ while neglected the fourth and higher power of $\xi$, which has a little impact on the electrical transports when the temperature is very close to the critical point. Secondly, we applied Taylor series expansion to the band-edge energy $E$ near $\xi_0$ based on the Landau theory rather than the quantum theory. Thirdly, the relationship between the Fermi wave vector $k_F$ and the Fermi energy $E_F$ was

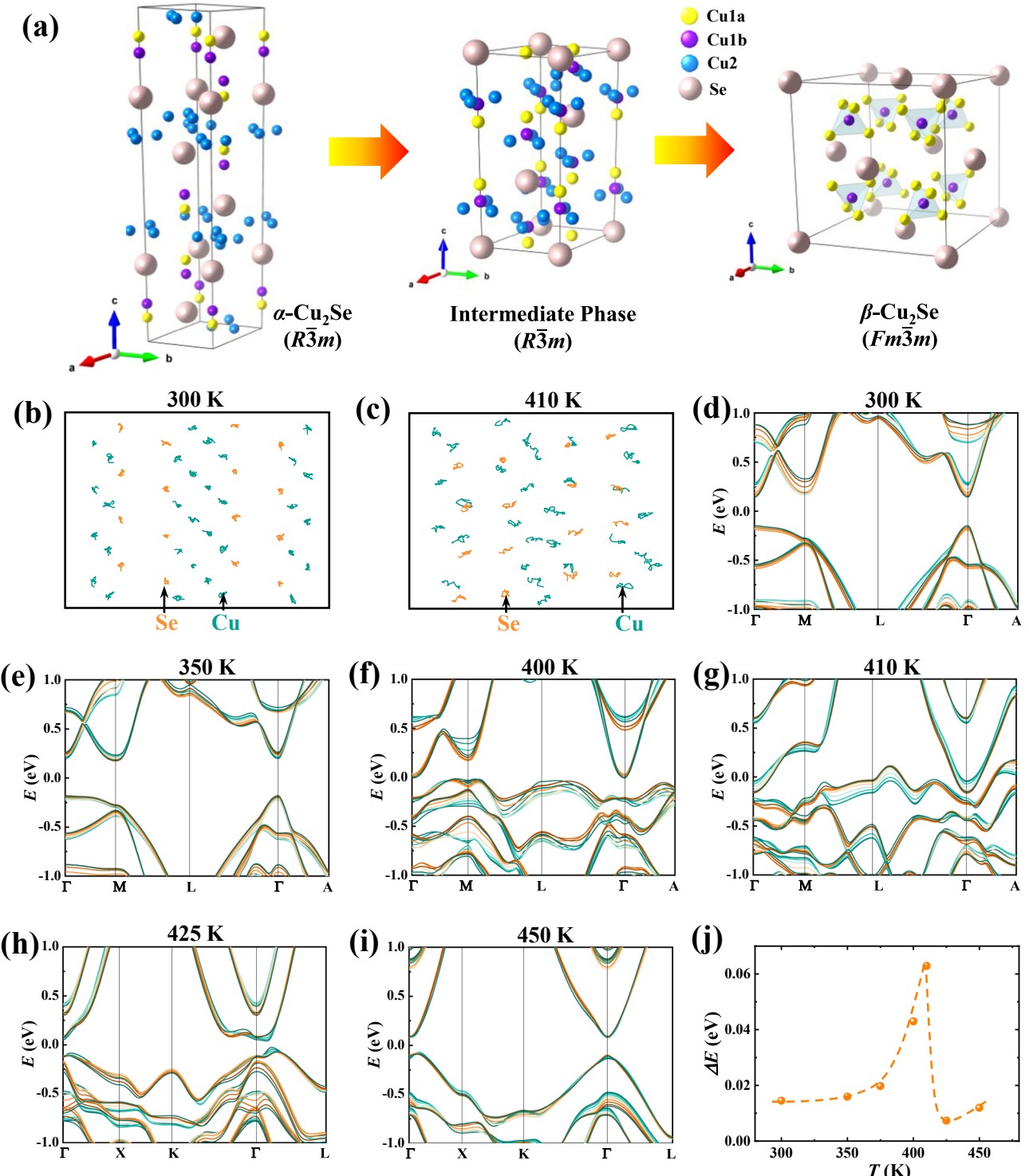

**Fig. 2 | Evolution of crystal structure and band structure during critical phase transitions for Cu₂Se. a** Crystal structures of the low temperature trigonal phase, intermediate phase, and high temperature cubic phase. Atomic trajectories derived from molecular dynamics simulations at (**b**) 300 K and (**c**) 410 K. The Cu and Se atoms are colored by cyan and yellow, respectively. Calculated band structures at (**d**) 300 K, (**e**) 350 K, (**f**) 400 K, (**g**) 410 K, (**h**) 425 K, and (**i**) 450 K. **j** The broadened energy $\Delta E$ of the valence band edge as a function of temperature.

derived using the single parabolic band model. The expressions of electrical resistivity $\rho_0$ and Seebeck coefficient $\alpha_0$ for the normal phases were also derived under the assumption of relaxation time approximation and single parabolic band of degenerate semiconductors. These approximations or assumptions may introduce certain biases or oversights between theory and experiment.

## Universality and verification of the model

The well-known liquid-like TE material Cu₂Se with critical phase transition is selected to verify our model. The phase transition of Cu₂Se is completely reversible except for slight hysteresis during cooling (Supplementary Fig. 1a). At room temperature, Cu₂Se adopts a layered trigonal structure $(R\bar{3}m)$[41] with all atoms orderly distributed

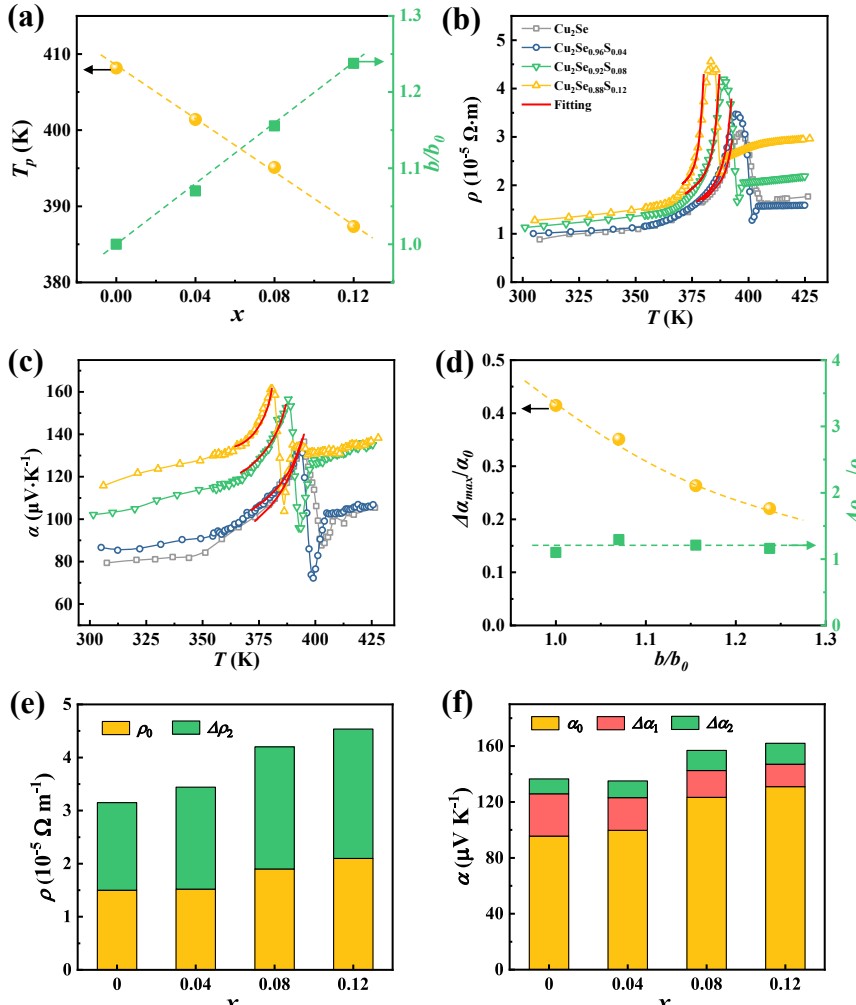

**Fig. 3 | Tuning the critical electrical transports during phase transitions in Cu₂Se₁₋ₓSₓ. a** Critical temperature $T_p$ and the relative phase transition parameter $b/b_0$ as a function of S content $x$. $b_0$ denotes the phase transition parameter for Cu₂Se. Temperature dependence of (**b**) electrical resistivity $\rho$ and (**c**) Seebeck coefficient $\alpha$. Red lines represent the fitting results by our model. **d** The maximum

relative increment of electrical resistivity $\Delta\rho_{max}/\rho_0$ and Seebeck coefficient $\Delta\alpha_{max}/\alpha_0$ at $T_p$ as a function of phase transition parameter $b/b_0$. The dashed lines are guide to the eyes. The respective contribution of (**e**) electrical resistivity and (**f**) Seebeck coefficient from the normal phase ($\rho_0$ and $\alpha_0$), the band broadening effect ($\Delta\alpha_1$), and the carrier-soft TO phonon interactions ($\Delta\rho_2$ and $\Delta\alpha_2$) at the critical point.

in the crystal lattice. Upon heating, part of Cu migrates from the copper rich to the copper deficient layer, ultimately transforming into a disordered cubic structure ($Fm\bar{3}m$)[42] above the critical temperature ($T_p$ ~ 410 K, see Fig. 2a). Accordingly, the order parameter goes continuously from finite value to zero at $T_p$. Ab initio molecular dynamics (MD) simulations and band structure calculations were performed to understand the structural fluctuation and band broadening effect. As shown in Fig. 2b, all atoms vibrate in the vicinity of their equilibrium positions at 300 K, whereas part of Cu atoms migrate toward the copper deficient layer and interact with Se from the adjacent neighboring layer at 410 K (Fig. 2c). Compared to the atomic trajectories at 300 K, both Cu and Se vibrate in a much larger space at 410 K, implying the crystal structure is undergoing a critical fluctuation near the critical temperature $T_p$. The valence band edge shows a clear broadening as $T$ approaches $T_p$. Specifically, the broadened energy $\Delta E$ is only 0.014 eV at 300 K, but it is sharply increased to 0.063 eV at 410 K and finally drops to 0.007 eV at 425 K (Fig. 2i). The $\lambda$-shaped $\Delta E$ implies that the band broadening is caused by structural fluctuations, not thermal vibrations. Besides, the band gap $E_g$ is obviously decreased as the temperature increases, which is probably caused by the enhanced interaction between Se and Cu.

The broadened band edge and decreased $E_g$ near $T_p$ well attest to the band broadening effect during critical phase transitions.

Step-wise increase trends in $\rho$ and $\alpha$ are observed in the normal trigonal and cubic phases of Cu₂Se. However, during the phase transition region, the $\rho$ sharply increases by a factor of 2.9 while $\alpha$ sharply increases by a factor of 1.4 when the temperature approaches to $T_p$. The obtained power factor ($PF = \alpha^2/\rho$) exhibits a trench-like shape near the critical point (Supplementary Fig. 2a). The sharp increasement of $\rho$ and $\alpha$ are obviously contributed by the carrier–soft transverse-optical (TO) phonon interaction and band broadening effect. Using the derived Eqs. (7) and (9), we fitted the critical electrical properties near $T_p$ with the fitting parameters shown in Table S1. Here the $b$ value is taken as $1.5 \times 10^{26}$ for both $\rho$ and $\alpha$. A good agreement between the experimental data (symbols in Fig. 3b, c) and theoretical estimation (red lines) is obtained in the phase transition region. This implies that our model is universal and can link all the critical electrical transports through parameter $b$. Furthermore, the fitted $T_p$ is 413 K, consistent well with the experimental data (409.5 K) determined from the measurement of $C_p$. And the fitted $\gamma$ value is on the order of $10^{25}$ s⁻², which is also comparable to the value of $7.6 \times 10^{25}$ s⁻² estimated from the neutron measurement[37,43]. All these results well support and verify our models.

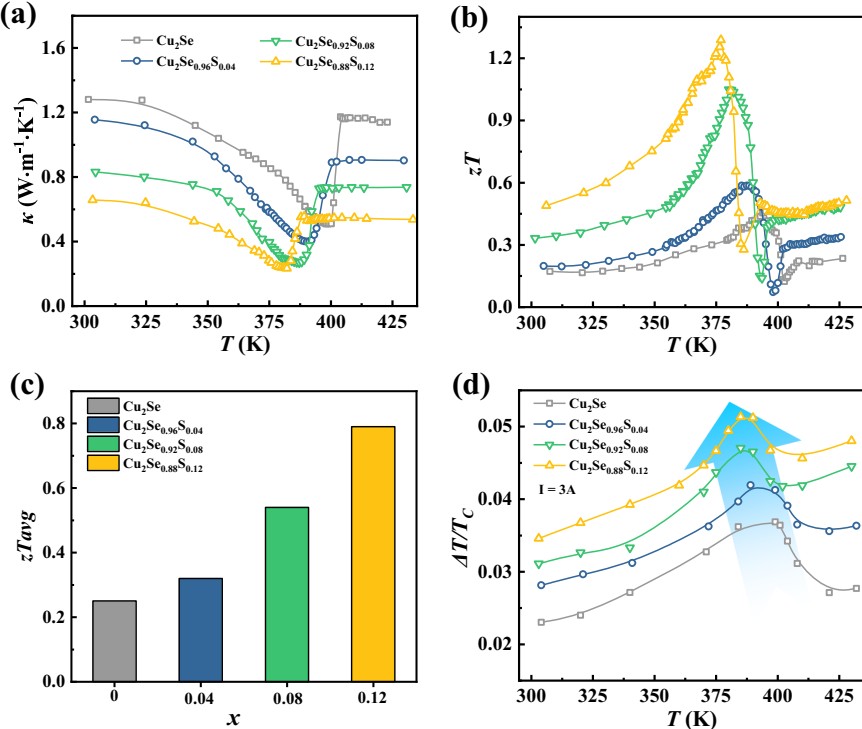

**Fig. 4 | Thermal conductivity, TE figure of merit *zT*, and cooling performance.**
Temperature dependence of (**a**) thermal conductivity $\kappa$ and (**b**) *zT* values for
$Cu_2Se_{1-x}S_x$ (*x* = 0, 0.04, 0.08 and 0.12). **c** Average *zT* from 300 K to the critical point.

**d** $\Delta T/T_C$ *vs* temperature measured at *I* = 3 A for the TE unicouple made of p-type
$Cu_2Se_{1-x}S_x$ and n-type $Yb_{0.3}Co_4Sb_{12}$. $T_C$ is the temperature of the cold junction side.
Solid lines are guides to the eyes.

## Tuning of the critical thermoelectric properties

Since the critical electrical transports during phase transitions are
directly affected by parameter *b* and phase transition temperature, we
try to alloy the analogous element S at Se sites in $Cu_2Se$ to tune the
features of critical phase transitions. The single trigonal phase with
homogenously distributed elements is observed (see XRD and EDS
results in Supplementary Fig. 3) in $Cu_2Se_{1-x}S_x$ solid solutions, with the
phase transition features greatly altered. The critical temperature ($T_p$)
is gradually reduced when increasing the content of S (Supplementary
Figs. 1b, 3a), which is mainly attributed to the less energy difference
between the low-temperature phase and high-temperature phase, i.e.
the smaller enthalpy change ($\Delta H$, Table S2) during phase transition for
S-alloyed $Cu_2Se$ compared to pristine $Cu_2Se$. Similar phenomenon has
been observed in a lot of solid solutions, such as $Cu_7PSe_{6-x}Te_x$[44] and
$Cu_{2-x}Ag_xTe$[45]. Meanwhile, the phase transition region is gradually nar-
rowed after the substitution of S at Se sites. According to the Landau
theory, the specific heat ($C_p$) near critical point is in direct proportion
to the square of phase transition parameter, i.e. $C_p \propto b^2$ (see details in
the Supplementary Information)[35]. Therefore, the relative phase tran-
sition parameter $b/b_0$ for $Cu_2Se_{1-x}S_x$ can be readily derived from the
slope of tangent lines near $T_p$ (see dashed lines in Supplementary
Fig. 1b). Clearly, when increasing S content, *b* is gradually improved
(Fig. 3a). Specifically, the *b* value for *x* = 0.12 is 1.24 times of that
in $Cu_2Se$.

Alloying S at the Se sites can increase the bonding energy to fix Cu
atoms in the crystal lattice[46], which successfully suppresses the forma-
tion of Cu vacancies and gives rise to the reduced hole concentra-
tions (i.e., the improved Fermi level). As a result, both electrical
resistivity $\rho$ and Seebeck coefficient $\alpha$ are greatly improved with
increasing S content. Similar to the temperature dependency of $Cu_2Se$,
step-wise increase in the normal phases and sharp increase during
phase transitions are observed for $\rho$ and $\alpha$ in $Cu_2Se_{1-x}S_x$. Alloying S
shifts the peak values to lower temperature, being in accordance with
the DSC results. Likewise, we fitted the abnormal electrical properties

near $T_p$ using Eqs. (7) and (9). The derived *b* values gradually increase
with increasing S content, which aligns with our experimental results.
The parameter $C_1$ gradually decreases while $\gamma$ gradually increases with
an increase of S content, which is also reasonable since $C_1$ is in an
inverse relation to the Fermi level while $\gamma$ is positively correlated with
the Fermi level. Figure 3d exhibits the maximum increment of elec-
trical resistivity $\Delta\rho_{max}/\rho_0$ and Seebeck coefficient $\Delta\alpha_{max}/\alpha_0$ at $T_p$ as a
function of $b/b_0$. When increasing $b/b_0$, the $\Delta\rho_{max}/\rho_0$ is nearly
unchanged but $\Delta\alpha_{max}/\alpha_0$ is gradually decreased, which are also fully
consistent with our theoretical prediction. Based on the fitting results
of our model, we can also derive the respective contributions from the
band broadening effect ($\Delta\alpha_1$) and the carrier-soft TO phonon interac-
tions ($\Delta\rho_2$ and $\Delta\alpha_2$). As shown in Figs. 3e and 3f, the enhanced resis-
tivity $\Delta\rho_2$ is gradually increased owing to the increased Fermi level.
Meanwhile, the enhanced Seebeck coefficient $\Delta\alpha_1$ during the critical
phase transition is reduced whereas $\Delta\alpha_2$ is slightly improved because
the former is largely impacted by the parameter *b* while the latter is
primarily determined by the scattering mechanism. Overall, alloying S
in $Cu_2Se$ proves to be an effective way to tune the strength of band
broadening effect and carrier-soft TO phonon interactions.

The critical fluctuation during phase transitions also has a large
impact on thermal transports. A sharp, cusp-like dip near the critical
point is observed in $\kappa$ for $Cu_2Se_{1-x}S_x$ (Fig. 4a). The minimal $\kappa$ is only
0.24 W m⁻¹ K⁻¹ near $T_p$ for $Cu_2Se_{0.08}S_{0.12}$, which is only one third of that
in normal phases. The lattice thermal conductivity $\kappa_L$ shows the similar
dip trend with $\kappa$, and the $\kappa_L$ value approaches zero at the critical point
(Supplementary Fig. 2c). The anomalous reduction in $\kappa_L$ during phase
transitions is majorly attributed to the strong coupling of heat-carrying
acoustic phonons with soft optic phonons[20]. Figure 4b shows the
temperature dependence of figure of merit *zT* for $Cu_2Se_{1-x}S_x$. Owing to
the significantly enhanced $\alpha$ and decreased $\kappa$, the *zT* values are greatly
improved during phase transitions. A maximum *zT* of 1.3 is achieved at
377 K for $Cu_2Se_{0.88}S_{0.12}$, which is at least 1.5 times larger than those of
normal trigonal and cubic phases, and represents a 190% improvement

over that of $Cu_2Se$. Moreover, the average $zT$ from 300 K to the critical point is also as high as 0.8 (Fig. 4c), which is also among the top values of TE materials. Such remarkable TE performance near room temperature makes it highly competitive with the state-of-the-art TE materials like $Bi_2Te_3$[47] and $Mg_3Bi_2$[48].

To further confirm the significantly improved $zT$ values during phase transitions, we constructed a series of unicouples consisting of p-type $Cu_2Se_{1-x}S_x$ and n-type $Yb_{0.3}Co_4Sb_{12}$ TE materials with the cooling performance shown in Supplementary Fig. 4 (see details in the Experimental Section). When the direct current is applied, the temperature at cold junction ($T_C$) is dramatically reduced due to the Peltier effect. When increasing current, the temperature drop ($\Delta T$) is gradually increased and eventually saturated at around 12 A (Supplementary Fig. 4). The maximum temperature difference ($\Delta T$) values achieved in the phase transition region are higher than those of normal trigonal and cubic phases. Figure 4d shows the values of $\Delta T/T_C$ at different temperatures under a small current of 3 A. Obviously, the temperature dependent $\Delta T/T_C$ shows a similar $\lambda$ shape as that of $zT$ values. $\Delta T/T_C$ is greatly improved during phase transitions with a maximum enhancement of more than 48% as compared with the trigonal and cubic phases. In addition, the maximum $\Delta T/T_C$ is shifted to lower temperatures when increasing S content.

In summary, previous theoretical models for critical electrical transports were either qualitative or solely focused on a specific mechanism. Herein, we propose a quantitative model covering both electrical conductivity and Seebeck coefficient to explain the critical properties during phase transitions by including the band broadening effect and carrier-soft TO phonon interactions. This model is well validated by the experiment data in $Cu_2Se$ and has been successfully used to explore and tune the critical features of S alloyed $Cu_2Se$. Considering the universal behavior often exhibited by critical phenomena, whereby the same underlying mathematical and physical principles apply across various systems, we believe our findings can not only offer valuable insights for future studies on critical thermoelectric transports but also provide inspiration for the exploration of other complex systems such as magnets, fluids, and biological systems. For example, the heat capacity anomaly at the critical transition is also caused by the fluctuations in the order parameter of the system, similar to the electrical properties in our model. At temperatures below the critical point, the order parameter is nonzero and the system is ordered. As the temperature approaches the critical point, the fluctuations in the order parameter become more pronounced and the system becomes more disordered. Therefore, theoretically, it is also possible to derive a model for heat capacity based on the Landau theory. Experimentally, the abnormal heat capacity during critical transitions have been observed in various systems, such as ferromagnetic $UGe_2$[5], liquid crystals[49], and $Pb_{1-x}Ge_xTe$ alloys[50]. This strongly suggests that the critical phenomena and its physical models are universal. It is worth noting that the narrow temperature range of phase transitions restricts its practical applications. Future studies should focus more on the tuning and broadening of phase transition temperature range. For example, we can alloy or dope the material with appropriate elements to reduce the energy difference between the low-temperature and high-temperature phases. Besides, by fabricating a functionally graded material with a multilayer structure, we can potentially harness the critical properties of different compositions and thus extend the temperature range of phase transition.

## Methods
### Materials synthesis
Polycrystalline $Cu_2Se_{1-x}S_x$ ($x = 0, 0.04, 0.08$ and $0.12$) and $Yb_{0.3}Co_4Sb_{12}$ were prepared from high purity elements: Cu (shots, 99.999%, Alfa Aesar), Se (shots, 99.999%, Alfa Aesar), S (pieces, 99.999%, Alfa Aesar), Co (shots, 99.99%, Alfa Aesar), Sb (shots, 99.999%, Alfa Aesar) and Yb (ingots, 99.95%, Alfa Aesar). The required elements were weighted out

in the pyrolytic nitride crucible according to the nominal chemical compositions. The crucible was enclosed in fused silica tube under vacuum. Then, the tubes were heated up to 1423 K in 12 h and kept at this temperature for 12 h, and then cooled to 923 K over 50 h. After being annealed at 923 K for 6 days, the tubes were cooled to ambient temperature naturally. Only a few cooper whiskers were observed on the surface of $Cu_2Se$, which were removed after annealing. Subsequently, the ingots were grounded into fine powder and sintered by a spark plasma sintering facility (SPS, Sumitomo SPS-2040) at 723 K for 10 min under a pressure of 65 MPa. Thermally conducting and electrically insulating BN layers were sprayed onto the inner surface of carbon dies and carbon foils to prevent the Cu migration during sintering process. The raw materials for $Yb_{0.3}Co_4Sb_{12}$ were sealed in the evacuated silica tubes, melted at 1353 K, and then quenched to room temperature, followed by annealing at 923 K for over 6 days. The grounded powder was sintered by SPS at 873 K under a pressure of 60 MPa. The relative density of all samples was above 98%.

### Thermoelectric unicouple fabrication
A series of thermoelectric unicouples based on p-type $Cu_2Se_{1-x}S_x$ and n-type $Yb_{0.3}Co_4Sb_{12}$ was constructed and tested. Firstly, p-type $Cu_2Se_{1-x}S_x$ and n-type $Yb_{0.3}Co_4Sb_{12}$ materials were diced into the desired dimensions ($A_p = 3 \times 3$ mm², $A_n = 1.67 \times 1.67$ mm², and $l_p = l_n = 1$ mm). Then, the obtained legs were treated by electroplating under a current density 100 A m⁻² to obtain low resistance ohmic contact (Supplementary Fig. 5a). After polishing the surrounding electroplated nickel, the p- and n-type legs were assembled by a soldering process, wherein the hot side was bridged by a copper-clad plate and the cold side was welded to a direct bonding copper substrate using a tin-based solder. To obtain the temperatures of cold and hot sides, two pairs of K-type thermocouples were soldered on the cold and hot junctions of the unicouple.

### Characterization
The room temperature powder X-ray diffraction (PXRD) were performed on a Rigaku, Rint 2000 instrument using Cu Kα source ($\lambda_{Cu-K\alpha} = 1.5405$ Å). The scan speed is 2°/min, and the scan step is 0.02°. The element distribution is investigated using field emission scanning electron microscopy (SEM, ZEISS SUPRA 55) equipped with energy dispersive microscopy (EDS, OXFORD Aztec XMax80). The differential scanning calorimetry (DSC) measurements were performed on Netzsch 200F3 with aluminum pans. To calculate the speed of phase transition, all samples were subjected to heating rates of 1.0, 2.0, 5.0, and 10.0 K min⁻¹. The thermal diffusivity ($\lambda_m$) was measured by laser flash method (Netzsch LFA457) under continuous argon flow. The temperature interval was set to be as small as 1 K during the phase transition region. The sample density ($d$) was measured by the Archimedes method. As mentioned previously, the thermal conductivity during the phase transition is calculated by $\kappa = C_{p0} \times d \times \lambda_0$, where $C_{p0}$, $\lambda_0$ are the heat capacity and thermal diffusivity without phase transition contribution, respectively. The total heat capacity $C_p$ could be divided into two parts: one is the normal heat capacity ($C_{p0}$) from phonons and electrons without contribution from phase transition, the other is the extra energy ($C_{pt}$) required to transform the low-temperature trigonal phase to high-temperature cubic phase. Therefore, by subtracting the $C_{pt}$ from $C_p$, we can obtain the value of $C_{p0}$, which is also close to the theoretical Dulong–Petit value. The Seebeck coefficient ($\alpha$) and electrical resistivity ($\rho$) during the phase transition were measured using our previously developed system based on a thermal expansion equipment (Netzsch DIL 402 C, Supplementary Fig. 5b)[6]. Two pairs of K-type thermocouples, two Ni electrodes, and one heater were packaged into the chamber of the thermal expansion system. The $\rho$ was measured using the standard four-probe method. The $\alpha$ was calculated from the slope of the $\Delta V$ vs. $\Delta T$ curve, with $\Delta T$ kept less than 3 K. The temperature dependent $\alpha$ and $\rho$ were measured

under a very small temperature step (around 1 K) and a very slow heating rate (around 0.1 K/min) during phase transitions. Moreover, each temperature point was maintained for more than 20 min to ensure consistency between the sample temperature and the ambient temperature. The electrical properties before and after the phase transitions were re-measured using a commercial apparatus ZEM-3 (ULVAC) to authenticate the data (Supplementary Fig. 5c and 5d).

The cooling performance of $Cu_2Se_{1-x}S_x/Yb_{0.3}Co_4Sb_{12}$ unicouple was characterized using a self-built measurement system. As shown in Supplementary Fig. 4a, the unicouple was placed on a large Cu heat sink with heating rods inside which enables the measurement at different hot side temperatures. The testing of cooling performance is performed in a vacuumed chamber under a pressure of $-10^{-3}$ Pa. During the measurement of $\Delta T$ with zero heat load, the DC current was provided by Agilent E3633A, the current and temperature of cold and hot sides were recorded by Keithley 2701 data acquisition system. Supplementary Fig. 4c shows the origin data measured at $T_C = 401$ K for $Cu_2Se$. The bule circles represent the cold side temperature on the top of p-leg. To verify the cooling performance during the phase transition, several hot side temperatures were selected to cover the temperature range of phase transitions in $Cu_2Se_{1-x}S_x$.

### First-principles calculations
Ab initio molecular dynamics simulations were performed for 15 ps with a time step of 1 fs at 300 K, 350 K, 375 K, 400 K, 410 K, 425 K, and 450 K, and atomic trajectories were collected from the last 2 ps. Nine atomic configurations were picked out every 0.2 ps at each temperature based on the results of MD simulations. The band structures were calculated using density functional theory with projector-augmented-wave (PAW) pseudopotential through the Vienna ab initio Simulation Package (VASP)[51]. The generalized gradient approximation method of Perdew–Burke–Ernzerhof (PBE) and the Hubbard-type correction $U_{eff}$ (4 eV)[52] of the Cu 3d orbital was used to describe the exchange and correlation of electrons. A plane-wave energy cutoff of 520 eV and an energy convergence criterion of $10^{-4}$ eV for self-consistency were adopted. The molecular dynamics simulation was conducted under constant pressure to account for the volume variation induced by temperature. As shown in Supplementary Fig. 6, the unit cell volume for the normal phases (<350 K and >425 K) gradually increases with rising temperature, demonstrating a thermal expansion effect. However, during the phase transition region, the sample exhibits a negative thermal expansion phenomenon, which is fully in line with the experimental observations[41].

## Data availability
All data are available in the manuscript and in the supplementary materials.

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

## Acknowledgements

This work was supported by the National Natural Science Foundation of China (No. 52372209, 52002406, and 91963208), the Shanghai Pilot Program for Basic Research-Chinese Academy of Science, Shanghai Branch (JCYJ-SHFY-2022-002). We acknowledge the computation support from the Center for High Performance Computing at Shanghai Jiao Tong University.

## Author contributions

K.Z., H.C. and X.S. designed the project. Z.Y., K.Z. and P.Q. prepared the samples and carried out the transport measurements. H.C. performed the first-principles calculations. K.Z., and H.C. constructed the theoretical framework. H. W., T.D., J.L., Z.Y. and K.Z. analyzed the data. Z.Y. and K.Z. wrote the original manuscript. L.C., X.S. and K.Z. supervised the project. All the authors reviewed and edited the manuscript.

## Competing interests

The authors declare no competing interests.

## Additional information

**Peer review information** : *Nature Communications* thanks Yue-Xing Chen, Zhuangfei Zhang and Nguyen Hieu for their contribution to the peer review of this work. A peer review file is available.

