## [Peer Review File · Nature Communications]

Modeling Critical Thermoelectric Transports Driven by Band Broadening and Phonon SofteningREVIEWER COMMENTS

Reviewer #1 (Remarks to the Author):

Critical phenomena, especially thermoelectrical properties during phase transitions, remain elusive due to experimental and theoretical challenges. In this work, the authors developed a quantitative theory that incorporates band broadening and carrier-soft TO phonon interactions to model the electrical transports during critical phase transitions. The universality of the model was furtherly validated by experimental data of Cu₂Se. The maximum zT reached a high value of 1.3 at the critical point, demonstrating the great potential of phase transition thermoelectrics for heat pumps or power generators near room temperature. Overall, the manuscript is properly structured, and the results are also very interesting and convincing. I think it may be of interest to the readers of areas like critical transports, phase-change materials, thermoelectrics, and other material sciences. However, there are still several unclear points in the current manuscript. Hence, the manuscript still needs minor revisions.

1. Despite the high zT value at the critical point, the narrow temperature range of phase transition restricts its practical application. Have the authors considered any strategies to broaden the temperature range of phase transition?
2. Cu₂Se is a typical superionic material at high temperature, how is the material stability when going through the phase transition upon heating and cooling?
3. It is still unclear to me why the carrier concentration decreased after alloying sulfur in Cu₂Se.
4. How to design high-performance phase transition thermoelectrics? The authors should give some comments or perspectives on this point.
5. The electrical measurement details should be included in the manuscript since it was performed on a home-made equipment.

Reviewer #2 (Remarks to the Author):

Understanding the fundamental mechanism behind the anomalous critical properties during phase transitions is very important and also very challenging. In this manuscript, Zhao et al. developed a quantitative theory to model the critical electrical transports by simultaneously considering the band broadening effect and the interaction between charge carriers and soft TO phonons. It is found that the phase transition parameter b , which can be tuned by doping or alloying, is a key factor in governing the increment of Seebeck coefficient and electrical resistivity. This universal model is well validated by the experimental data of S alloyed Cu₂Se. In general, the paper is well organized and carefully written. The topic is also of considerable interest to the researchers in the related areas. I think this paper can be accepted after following comments being properly addressed:

1. Is the molecular dynamics simulation performed under constant pressure or constant volume? Does it take into account the volume expansion induced by temperature?
2. It is typically challenging to accurately measure the electrical resistivity and Seebeck coefficient during the phase transitions. The authors should provide more details regarding the measurements.
3. It is recommended to present the data of lattice thermal conductivity and power factor to enhance readers' understanding of the impact of critical phenomena on electrical and thermal transport performance.

Referee#1

Comments to the Author: *Critical phenomena, especially thermoelectrical properties during phase transitions, remain elusive due to experimental and theoretical challenges. In this work, the authors developed a quantitative theory that incorporates band broadening and carrier-soft TO phonon interactions to model the electrical transports during critical phase transitions. The universality of the model was furtherly validated by experimental data of Cu₂Se. The maximum zT reached a high value of 1.3 at the critical point, demonstrating the great potential of phase transition thermoelectrics for heat pumps or power generators near room temperature. Overall, the manuscript is properly structured, and the results are also very interesting and convincing. I think it may be of interest to the readers of areas like critical transports, phase-change materials, thermoelectrics, and other material sciences. However, there are still several unclear points in the current manuscript. Hence, the manuscript still needs minor revisions.*

Reply: We appreciate the reviewer's overall positive comments.

Comment 1: *Despite the high zT value at the critical point, the narrow temperature range of phase transition restricts its practical application. Have the authors considered any strategies to broaden the temperature range of phase transition?*

Reply: Thanks for the insightful comments and suggestions. It is indeed very important to broaden the temperature range of phase transition for real applications. The phase transition temperature can be tuned by doping the material with an appropriate element like Ag, I, or S.^{1,2} We can then leverage the critical properties of different compositions by fabricating a functionally graded material with multilayer structure. Experimentally, we have prepared a three-graded Cu₂Se_{1-x}S_x material with $x = 0.04, 0.08,$ and 0.12 and measured its electrical properties. As shown in **Figure R1**, the temperature range of phase transition is largely broadened, and better thermoelectric performance is achieved in the graded Cu₂Se_{1-x}S_x material. This strategy can be applied to the tuning of critical phenomenon for real applications.

Figure R1. (a) Schematic of the preparation of the graded $\text{Cu}_2\text{Se}_{1-x}\text{S}_x$ material. Temperature dependence of (b) electrical resistivity and (c) Seebeck coefficient for the graded $\text{Cu}_2\text{Se}_{1-x}\text{S}_x$ material during phase transition.

Comment 2: *Cu_2Se is a typical superionic material at high temperature, how is the material stability when going through the phase transition upon heating and cooling?*

Reply: We have measured the heat flow and electrical conductivity upon heating and cooling for Cu_2Se . The heat flow data demonstrate that the phase transition of Cu_2Se is completely reversible except for slight hysteresis during cooling (**Figure R2**). The temperature dependent electrical conductivity σ exhibits different behavior during cooling compared to the heating process due to the structural hysteresis effect (**Figure R3a**). After cooling, the room temperature σ slightly decreases from $1 \times 10^5 \text{ S m}^{-1}$ to $9.45 \times 10^4 \text{ S m}^{-1}$, and it recovers to $1 \times 10^5 \text{ S m}^{-1}$ after being kept at room temperature for 8 hours (**Figure R3b**).

Figure R2. Temperature dependence of heat flow during heating and cooling for Cu_2Se .

Figure R3. (a) Temperature dependence of electrical conductivity for Cu_2Se with two cycles during phase transition. (b) Time dependence of electrical conductivity for Cu_2Se at room temperature.

Comment 3: *It is still unclear to me why the carrier concentration decreased after alloying sulfur in Cu_2Se .*

Reply: The nature of Cu vacancies in $\text{Cu}_2\delta$ ($\delta = \text{S}, \text{Se}, \text{Te}$) makes them p-type semiconductors with holes as the dominant charge carriers.³ Low hole concentrations ($\sim 5 \times 10^{18} \text{ cm}^{-3}$) are observed in Cu_2S due to the strong chemical bonding between Cu and S, whereas high hole concentrations ($\sim 6 \times 10^{20} \text{ cm}^{-3}$) are detected in Cu_2Se due to the relatively weak bonding between Cu and Se. Alloying S at the Se sites of Cu_2Se can increase the bonding energy to fix Cu atoms in the crystal lattice, which successfully suppresses the formation of Cu vacancies and gives rise to the reduced carrier concentrations.³

Comment 4: *How to design high-performance phase transition thermoelectrics? The authors should give some comments or perspectives on this point.*

Reply: Based on our theoretical modelling and experimental data, high-performance phase transition thermoelectrics should have a small phase transition parameter b and/or a wide phase transition temperature range. To achieve this, we can alloy or dope the material with appropriate elements to reduce the energy difference between the low-temperature and high-temperature phases, thereby decreasing the b values. Additionally, by fabricating a functionally graded material with a multilayer structure, we can harness the critical properties of different compositions and extend the temperature range of phase transition. We have added several sentences in the revised manuscript to discuss this point.

Comment 5: *The electrical measurement details should be included in the manuscript since it was performed on a home-made equipment.*

Reply: We measured the Seebeck coefficient (α) and electrical resistivity (ρ) during the phase transition using our previously developed system based on a thermal expansion equipment (Netzsch DIL 402C, **Figure R4**).² Two pairs of K-type thermocouples, two Ni electrodes, and one heater were packaged into the chamber of the thermal expansion system. The ρ was measured using the standard four-probe method. The α was calculated from the slope of the ΔV vs. ΔT curve, with ΔT kept less than 3 K. The temperature dependent α and ρ were measured under a very small temperature step (around 1 K) and a very slow heating rate (around 0.1 K/min) during phase transitions. The electrical properties before and after the phase transitions were re-measured using a commercial apparatus ZEM-3 (ULVAC) to authenticate the data. We have added the electrical measurement details in the revised manuscript.

Figure R4. Schematic of the home-made equipment for measuring the Seebeck coefficient and electrical resistivity during the phase transition.

Referee#2

Comments to the Author: *Understanding the fundamental mechanism behind the anomalous critical properties during phase transitions is very important and also very challenging. In this manuscript, Zhao et al. developed a quantitative theory to model the critical electrical transports by simultaneously considering the band broadening effect and the interaction between charge carriers and soft TO phonons. It is found that the phase transition parameter b , which can be tuned by doping or alloying, is a key factor in governing the increment of Seebeck coefficient and electrical resistivity. This universal model is well validated by the experimental data of S alloyed Cu_2Se . In general, the paper is well organized and carefully written. The topic is also of considerable interest to the researchers in the related areas. I think this paper can be accepted after following comments being properly addressed:*

Reply: We appreciate the reviewer's overall positive comments.

Comment 1: *Is the molecular dynamics simulation performed under constant pressure or constant volume? Does it take into account the volume expansion induced by temperature?*

Reply: The molecular dynamics simulation was conducted under constant pressure to account for the volume variation induced by temperature. As shown in **Figure R5**, the unit cell volume for the normal phases (< 350 K and > 425 K) gradually increases with rising temperature, demonstrating a thermal expansion effect. However, during the phase transition region, the sample exhibits a negative thermal expansion phenomenon, which is fully in line with the experimental observations.⁴ We have added these results in the revised manuscript.

Figure R5. The unit cell volume as a function of temperature.

Comment 2: *It is typically challenging to accurately measure the electrical resistivity and Seebeck coefficient during the phase transitions. The authors should provide more details regarding the measurements.*

Reply: We measured the Seebeck coefficient (α) and electrical resistivity (ρ) during the phase transition using our previously developed system based on a thermal expansion equipment (Netzsch DIL 402C, **Figure R4**).² Two pairs of K-type thermocouples, two Ni electrodes, and one heater were packaged into the chamber of the thermal expansion system. The ρ was measured using the standard four-probe method. The α was calculated from the slope of the ΔV vs. ΔT curve, with ΔT kept less than 3 K. The temperature dependent α and ρ were measured under a very small temperature step (around 1 K) and a very slow heating rate (around 0.1 K/min) during phase transitions. The electrical properties before and after the phase transitions were re-measured using a commercial apparatus ZEM-3 (ULVAC) to authenticate the data. We have added the electrical measurement details in the revised manuscript.

Comment 3: *It is recommended to present the data of lattice thermal conductivity and power factor to enhance readers' understanding of the impact of critical phenomena on electrical and thermal transport performance.*

Reply: Thanks for the suggestions. We have calculated the temperature-dependent lattice thermal conductivity and power factor (**Figure R6**) and included the data in the revised manuscript. Several sentences have been also added in the revised manuscript to discuss the lattice thermal conductivity and power factor.

Figure R6. Temperature dependence of (a) lattice thermal conductivity κ_L and (b) power factor PF for $\text{Cu}_2\text{Se}_{1-x}\text{S}_x$.

References

1. Brown DR, Day T, Borup KA, Christensen S, Iversen BB, Snyder GJ. Phase transition enhanced thermoelectric figure-of-merit in copper chalcogenides. *APL Mater* **1**, 052107 (2013).
2. Liu H, *et al.* Ultrahigh Thermoelectric Performance by Electron and Phonon Critical Scattering in $\text{Cu}_2\text{Se}_{1-x}\text{I}_x$. *Adv Mater* **25**, 6607-6612 (2013).
3. Zhao K, *et al.* Enhanced Thermoelectric Performance through Tuning Bonding Energy in $\text{Cu}_2\text{Se}_{1-x}\text{S}_x$ Liquid-like Materials. *Chem Mater* **29**, 6367-6377 (2017).
4. Eikeland E, *et al.* Crystal structure across the β to α phase transition in thermoelectric Cu_{2-x}Se . *IUCrJ* **4**, 476-485 (2017).

Summary of the changes

1. One sentence is added on page 10 and one figure (i.e. Fig. S1a) is added on page 7 of the Supplementary Information to demonstrate the reversible phase transition of Cu_2Se .
2. One sentence is added in 2nd paragraph of page 11 to discuss the power factor.
3. Two sentences are added in the 2nd paragraph of page 5 to explain the reduced hole concentration in S alloyed Cu_2Se .
4. One sentence is added in 1st paragraph of page 15 to discuss the lattice thermal conductivity.
5. Fig. 4 is restructured and the corresponding caption is modified on page 16.
6. Four sentences are added in the 2nd paragraphs on page 17 to prospect the strategies for tuning and broadening phase transition temperature.
7. The electrical measurement details are added in the Methods section.

8. Three sentences are added in the Methods section and one figure (i.e. Fig. S6) is add in the Supplementary Information to discuss the variation of unit cell volume.
9. Two figures including the power factor and lattice thermal conductivity are added in the Fig. S2 of the Supplementary Information.
10. The schematic of the home-made equipment is added in Fig. S5.

We sincerely thank the two reviewers again for their careful consideration and very helpful suggestions as well as comments. We believe that we have made clear explanations and proper modifications.

Sincerely yours,

Xun Shi

Professor Xun Shi

Shanghai Institute of Ceramics, Chinese Academy of Sciences,

and School of Material Science and Engineering, Shanghai Jiao Tong University

Tel: +86-138-1871-0650

E-mail: xshi@mail.sic.ac.cn (or xshi@sjtu.edu.cn)

Reviewer #3 (Remarks to the Author):

The paper addresses the challenging area of critical phenomena in modern physics, specifically focusing on anomalous thermoelectric properties during critical phase transitions. The authors highlight that existing theoretical models for critical electrical transports are either qualitative or limited to specific transport parameters. From that, the paper introduces a quantitative theory incorporating both the band broadening effect and carrier-soft TO (transverse optical) phonon interactions to model electrical transports during critical phase transitions. The band-broadening effect contributes to an additional term in the Seebeck coefficient, while carrier-soft TO phonon interactions significantly impact both electrical resistivity and the Seebeck coefficient.

The proposed model's universality and validity are asserted through confirmation with experimental data. Additionally, the authors explore the tunability of critical phase transition features, illustrating this with the example of alloying S (sulfur) in Cu₂Se (copper selenide), which affects phase transition temperature and parameter b . The paper claims a notable achievement in pushing the maximum thermoelectric figure of merit zT to a record value of 1.3 at the critical point (377 K), surpassing values observed in normal static phases.

The significance of the work is emphasized by its provision of a clearer understanding of critical electrical transports and the presentation of new guidelines for future studies in this area. The manuscript is well written, but some explanations and clarifications would help to better understand the underlying assumptions and the implications of the presented materials.

1. How carefully was the methodology used to create the quantitative model for electrical transports during critical phase transitions, and are there any possible biases or oversights in the approach?
2. How applicable is the proposed quantitative model to other systems experiencing critical phase transitions, and does the paper talk about any limitations or conditions that might restrict its use in different situations?
3. The paper claims the universality and validity of the model through experimental data. How thorough and independent is the experimental validation process, and does it sufficiently consider potential confounding variables?
4. Does the model have limitations in capturing certain aspects of critical phase transitions, and are there areas where it may need further refinement?
5. The paper talks about adjusting critical features like phase transition temperature and parameter b by adding S to Cu₂Se. How strong is the evidence supporting these findings, and are there other possible explanations for the observed effects of alloying?
6. The paper highlights achieving a remarkable thermoelectric figure of merit, zT , at the critical point. What specific conditions or characteristics of the critical phase

transition contribute to this outstanding value, and how consistently does this enhancement persist under different conditions?

7. The paper proposes that its findings might offer insights into complex systems like magnets, fluids, and biological systems due to the universal characteristics of critical phenomena. How extensively does the paper examine the practicality and transferability of these findings across different scientific areas?
8. In Eq. (2), why do the authors use the classical Landau theory for the distribution function of the band energy E , instead of the quantum theory?

Referee#1

Comments to the Author: *Critical phenomena, especially thermoelectrical properties during phase transitions, remain elusive due to experimental and theoretical challenges. In this work, the authors developed a quantitative theory that incorporates band broadening and carrier-soft TO phonon interactions to model the electrical transports during critical phase transitions. The universality of the model was furtherly validated by experimental data of Cu₂Se. The maximum zT reached a high value of 1.3 at the critical point, demonstrating the great potential of phase transition thermoelectrics for heat pumps or power generators near room temperature. Overall, the manuscript is properly structured, and the results are also very interesting and convincing. I think it may be of interest to the readers of areas like critical transports, phase-change materials, thermoelectrics, and other material sciences. However, there are still several unclear points in the current manuscript. Hence, the manuscript still needs minor revisions.*

Reply: We appreciate the reviewer's overall positive comments.

Comment 1: *Despite the high zT value at the critical point, the narrow temperature range of phase transition restricts its practical application. Have the authors considered any strategies to broaden the temperature range of phase transition?*

Reply: Thanks for the insightful comments and suggestions. It is indeed very important to broaden the temperature range of phase transition for real applications. The phase transition temperature can be tuned by doping the material with an appropriate element like Ag, I, or S.^{1,2} We can then leverage the critical properties of different compositions by fabricating a functionally graded material with multilayer structure. Experimentally, we have prepared a three-graded Cu₂Se_{1-x}S_x material with $x = 0.04, 0.08,$ and 0.12 and measured its electrical properties. As shown in **Figure R1**, the temperature range of phase transition is largely broadened, and better thermoelectric performance is achieved in the graded Cu₂Se_{1-x}S_x material. This strategy can be applied to the tuning of critical phenomenon for real applications.

Figure R1. (a) Schematic of the preparation of the graded $\text{Cu}_2\text{Se}_{1-x}\text{S}_x$ material. Temperature dependence of (b) electrical resistivity and (c) Seebeck coefficient for the graded $\text{Cu}_2\text{Se}_{1-x}\text{S}_x$ material during phase transition.

Comment 2: *Cu_2Se is a typical superionic material at high temperature, how is the material stability when going through the phase transition upon heating and cooling?*

Reply: We have measured the heat flow and electrical conductivity upon heating and cooling for Cu_2Se . The heat flow data demonstrate that the phase transition of Cu_2Se is completely reversible except for slight hysteresis during cooling (**Figure R2**). The temperature dependent electrical conductivity σ exhibits different behavior during cooling compared to the heating process due to the structural hysteresis effect (**Figure R3a**). After cooling, the room temperature σ slightly decreases from $1 \times 10^5 \text{ S m}^{-1}$ to $9.45 \times 10^4 \text{ S m}^{-1}$, and it recovers to $1 \times 10^5 \text{ S m}^{-1}$ after being kept at room temperature for 8 hours (**Figure R3b**).

Figure R2. Temperature dependence of heat flow during heating and cooling for Cu_2Se .

Figure R3. (a) Temperature dependence of electrical conductivity for Cu_2Se with two cycles during phase transition. (b) Time dependence of electrical conductivity for Cu_2Se at room temperature.

Comment 3: *It is still unclear to me why the carrier concentration decreased after alloying sulfur in Cu_2Se .*

Reply: The nature of Cu vacancies in $\text{Cu}_2\delta$ ($\delta = \text{S}, \text{Se}, \text{Te}$) makes them p-type semiconductors with holes as the dominant charge carriers.³ Low hole concentrations ($\sim 5 \times 10^{18} \text{ cm}^{-3}$) are observed in Cu_2S due to the strong chemical bonding between Cu and S, whereas high hole concentrations ($\sim 6 \times 10^{20} \text{ cm}^{-3}$) are detected in Cu_2Se due to the relatively weak bonding between Cu and Se. Alloying S at the Se sites of Cu_2Se can increase the bonding energy to fix Cu atoms in the crystal lattice, which successfully suppresses the formation of Cu vacancies and gives rise to the reduced carrier concentrations.³

Comment 4: *How to design high-performance phase transition thermoelectrics? The authors should give some comments or perspectives on this point.*

Reply: Based on our theoretical modelling and experimental data, high-performance phase transition thermoelectrics should have a small phase transition parameter b and/or a wide phase transition temperature range. To achieve this, we can alloy or dope the material with appropriate elements to reduce the energy difference between the low-temperature and high-temperature phases, thereby decreasing the b values. Additionally, by fabricating a functionally graded material with a multilayer structure, we can harness the critical properties of different compositions and extend the temperature range of phase transition. We have added several sentences in the revised manuscript to discuss this point.

Comment 5: *The electrical measurement details should be included in the manuscript since it was performed on a home-made equipment.*

Reply: We measured the Seebeck coefficient (α) and electrical resistivity (ρ) during the phase transition using our previously developed system based on a thermal expansion equipment (Netzsch DIL 402C, **Figure R4**).² Two pairs of K-type thermocouples, two Ni electrodes, and one heater were packaged into the chamber of the thermal expansion system. The ρ was measured using the standard four-probe method. The α was calculated from the slope of the ΔV vs. ΔT curve, with ΔT kept less than 3 K. The temperature dependent α and ρ were measured under a very small temperature step (around 1 K) and a very slow heating rate (around 0.1 K/min) during phase transitions. The electrical properties before and after the phase transitions were re-measured using a commercial apparatus ZEM-3 (ULVAC) to authenticate the data. We have added the electrical measurement details in the revised manuscript.

Figure R4. Schematic of the home-made equipment for measuring the Seebeck coefficient and electrical resistivity during the phase transition.

Referee#2

Comments to the Author: *Understanding the fundamental mechanism behind the anomalous critical properties during phase transitions is very important and also very challenging. In this manuscript, Zhao et al. developed a quantitative theory to model the critical electrical transports by simultaneously considering the band broadening effect and the interaction between charge carriers and soft TO phonons. It is found that the phase transition parameter b , which can be tuned by doping or alloying, is a key factor in governing the increment of Seebeck coefficient and electrical resistivity. This universal model is well validated by the experimental data of S alloyed Cu_2Se . In general, the paper is well organized and carefully written. The topic is also of considerable interest to the researchers in the related areas. I think this paper can be accepted after following comments being properly addressed:*

Reply: We appreciate the reviewer's overall positive comments.

Comment 1: *Is the molecular dynamics simulation performed under constant pressure or constant volume? Does it take into account the volume expansion induced by temperature?*

Reply: The molecular dynamics simulation was conducted under constant pressure to account for the volume variation induced by temperature. As shown in **Figure R5**, the unit cell volume for the normal phases (< 350 K and > 425 K) gradually increases with rising temperature, demonstrating a thermal expansion effect. However, during the phase transition region, the sample exhibits a negative thermal expansion phenomenon, which is fully in line with the experimental observations.⁴ We have added these results in the revised manuscript.

Figure R5. The unit cell volume as a function of temperature.

Comment 2: *It is typically challenging to accurately measure the electrical resistivity and Seebeck coefficient during the phase transitions. The authors should provide more details regarding the measurements.*

Reply: We measured the Seebeck coefficient (α) and electrical resistivity (ρ) during the phase transition using our previously developed system based on a thermal expansion equipment (Netzsch DIL 402C, **Figure R4**).² Two pairs of K-type thermocouples, two Ni electrodes, and one heater were packaged into the chamber of the thermal expansion system. The ρ was measured using the standard four-probe method. The α was calculated from the slope of the ΔV vs. ΔT curve, with ΔT kept less than 3 K. The temperature dependent α and ρ were measured under a very small temperature step (around 1 K) and a very slow heating rate (around 0.1 K/min) during phase transitions. The electrical properties before and after the phase transitions were re-measured using a commercial apparatus ZEM-3 (ULVAC) to authenticate the data. We have added the electrical measurement details in the revised manuscript.

Comment 3: *It is recommended to present the data of lattice thermal conductivity and power factor to enhance readers' understanding of the impact of critical phenomena on electrical and thermal transport performance.*

Reply: Thanks for the suggestions. We have calculated the temperature-dependent lattice thermal conductivity and power factor (**Figure R6**) and included the data in the revised manuscript. Several sentences have been also added in the revised manuscript to discuss the lattice thermal conductivity and power factor.

Figure R6. Temperature dependence of (a) lattice thermal conductivity κ_L and (b) power factor PF for $\text{Cu}_2\text{Se}_{1-x}\text{S}_x$.

Referee#3

Comments to the Author: *The paper addresses the challenging area of critical phenomena in modern physics, specifically focusing on anomalous thermoelectric properties during critical phase transitions. The authors highlight that existing theoretical models for critical electrical transports are either qualitative or limited to specific transport parameters. From that, the paper introduces a quantitative theory incorporating both the band broadening effect and carrier-soft TO (transverse optical) phonon interactions to model electrical transports during critical phase transitions. The band-broadening effect contributes to an additional term in the Seebeck coefficient, while carrier-soft TO phonon interactions significantly impact both electrical resistivity and the Seebeck coefficient.*

The proposed model's universality and validity are asserted through confirmation with experimental data. Additionally, the authors explore the tunability of critical phase transition features, illustrating this with the example of alloying S (sulfur) in Cu₂Se (copper selenide), which affects phase transition temperature and parameter b . The paper claims a notable achievement in pushing the maximum thermoelectric figure of merit zT to a record value of 1.3 at the critical point (377 K), surpassing values observed in normal static phases. The significance of the work is emphasized by its provision of a clearer understanding of critical electrical transports and the presentation of new guidelines for future studies in this area. The

manuscript is well written, but some explanations and clarifications would help to better understand the underlying assumptions and the implications of the presented materials.

Reply: We appreciate the reviewer's overall positive comments.

Comment 1: *How carefully was the methodology used to create the quantitative model for electrical transports during critical phase transitions, and are there any possible biases or oversights in the approach?*

Reply: This is a good question. We developed the quantitative model based on the Landau theory and the Boltzmann transport equation. There are certain small biases or oversights in the approach. Firstly, we only considered the second power of the order parameter ξ while neglected the fourth power of ξ , which has a little impact on the electrical transports when the temperature is very close to the critical point. Secondly, we applied Taylor series expansion to the band-edge energy E near ξ_0 based on the Landau theory rather than the quantum theory. Thirdly, the relationship between the Fermi wave vector k_F and the Fermi energy E_F (Eq. S13) was derived using the single parabolic band model. The expressions of electrical resistivity ρ_0 and Seebeck coefficient α_0 for the normal phases were also derived under the assumption of relaxation time approximation and single parabolic band of degenerate semiconductors. These approximations or assumptions may introduce certain biases or oversights between theory and experiment. We have added several sentences in the revised manuscript to discuss this point.

Comment 2: *How applicable is the proposed quantitative model to other systems experiencing critical phase transitions, and does the paper talk about any limitations or conditions that might restrict its use in different situations?*

Reply: The quantitative model was built upon the classical Landau theory and Boltzmann transport equation that are applicable to other systems. Some expressions in the model were derived under the assumption of relaxation time approximation and single parabolic band. Therefore, our model is applicable to the systems with single type carrier and parabolic energy band, such as RbAg_4I_5 ⁵, GeTe ⁶, and SnTe ^{7,8}. Indeed, similar critical phenomena have been observed in these material systems. We have added the limitations and conditions that restrict its use in the revised manuscript.

Comment 3: *The paper claims the universality and validity of the model through experimental data. How thorough and independent is the experimental validation process, and does it sufficiently consider potential confounding variables?*

Reply: Thanks for the insightful comments. The well-known thermoelectric material Cu_2Se with critical phase transition was selected to verify our model. The experimental processes were totally independent on the modelling. In order to accurately measure the Seebeck coefficient (α) and electrical resistivity (ρ) during the phase transition, we developed a new system based on the thermal expansion equipment (Netzsch DIL 402C, **Figure R4**).² We considered the impact of potential variables and factors during the measurement. The ρ was measured using the standard four-probe method. The α was calculated from the slope of the ΔV vs. ΔT curve, with ΔT kept within a small interval (< 3 K). The temperature dependent α and ρ were measured under a very small temperature step (around 1 K) and a very slow heating rate (around 0.1 K/min) during phase transitions. Moreover, each temperature point was maintained for more than 20 minutes to ensure consistency between the sample temperature and the ambient temperature. The electrical properties before and after the phase transitions were re-measured using a commercial apparatus ZEM-3 (ULVAC) to authenticate the data. Therefore, the experiment measurements and data in this work are independent and sufficient for the critical thermoelectric transports. We have added these description into the revised manuscript.

Comment 4: *Does the model have limitations in capturing certain aspects of critical phase transitions, and are there areas where it may need further refinement?*

Reply: Please see our response to comment 1. There are some approximations or assumptions in our model, which may lead to certain biases or oversights between theory and experiment. Besides, we have only built the model for the electrical transports during critical phase transitions. Further investigations are needed to model the thermal transports. In the future, once these details are included or can be better modeled, our model can be further refined or improved.

Comment 5: *The paper talks about adjusting critical features like phase transition temperature and parameter b by adding S to Cu_2Se . How strong is the evidence supporting these findings, and are there other possible explanations for the observed effects of alloying?*

Reply: This is a good question. We carefully measured the heat capacity C_p for all $\text{Cu}_2\text{Se}_{1-x}\text{S}_x$ samples using the same test conditions by differential scanning calorimetry (DSC). The temperature dependent C_p

data (Fig. S1) clearly showed that the phase transition temperature T_p was gradually reduced when increasing the content of S. The relative phase transition parameter b/b_0 was derived from the slope of tangent lines of C_p data. It is also evident that b was gradually improved with increasing S content.

The reduced T_p is mainly attributed to the less energy difference between the low-temperature phase and high-temperature phase, i.e. the smaller enthalpy change (ΔH) during phase transition for S-alloyed Cu_2Se compared to pristine Cu_2Se . Similar phenomenon has been observed in a lot of solid solutions, such as $\text{Cu}_7\text{PSe}_{6-x}\text{Te}_x$ ⁹ and $\text{Cu}_{2-x}\text{Ag}_x\text{Te}$ ¹⁰. The initial temperature of phase transition is similar for all samples, while the finish temperature of S-alloyed sample is lower than that of pristine Cu_2Se , leading to a narrow interval of phase transition in $\text{Cu}_2\text{Se}_{1-x}\text{S}_x$ solid solutions. Besides, due to the reduction of T_p , higher energy is required for the fluctuation of ζ . Consequently, the phase transition parameter b is reduced after alloying S in Cu_2Se .

Comment 6: *The paper highlights achieving a remarkable thermoelectric figure of merit, zT , at the critical point. What specific conditions or characteristics of the critical phase transition contribute to this outstanding value, and how consistently does this enhancement persist under different conditions?*

Reply: The thermoelectric figure of merit is given by $zT = S^2\sigma/\kappa T$, where S , σ , κ , T are the Seebeck coefficient, electrical conductivity, total thermal conductivity, and absolute temperature, respectively. Herein, the remarkable zT value is attributed to the enhanced Seebeck coefficient driven by the band broadening effect and carrier-soft TO phonon interactions, the reduced thermal conductivity caused by the softening of phonons, as well as the optimized carrier concentration achieved by S alloying. The enhanced resistivity is unfavorable for the thermoelectric performance. It is worth noting that the band-broadening effect contributes an additional term to Seebeck coefficient but has no impact on the resistivity. Therefore, strengthening the band broadening effect is beneficial for further improving the zT value. For critical thermoelectric materials, the Seebeck coefficient is greatly enhanced and the thermal conductivity is suppressed, while electrical conductivity is largely reduced. If the band-broadening effect is more obvious, the zT is larger. We have added a few sentences into the revised manuscript.

Comment 7: *The paper proposes that its findings might offer insights into complex systems like magnets, fluids, and biological systems due to the universal characteristics of critical phenomena. How extensively does the paper examine the practicality and transferability of these findings across different scientific areas?*

Reply: Critical phenomena are observed in a wide range of physical systems, such as magnets, fluids, and alloys, when they undergo a critical phase transition. The behavior of a system at a critical point is characterized by several universal properties, such as critical fluctuation of order parameters, scaling relations among different quantities, and power-law divergences of some quantities, which are independent of the specific system or microscopic details. Therefore, we believe our findings may provide inspiration for the exploration of other complex systems experiencing critical transitions.

For example, the heat capacity anomaly at the critical transition is also caused by the fluctuations in the order parameter of the system, similar to the electrical properties in our model. At temperatures below the critical point, the order parameter is nonzero and the system is ordered. As the temperature approaches the critical point, the fluctuations in the order parameter become more pronounced and the system becomes more disordered. Therefore, theoretically, it is also possible to derive a model for heat capacity based on the Landau theory. Experimentally, the abnormal heat capacity during critical transitions have been observed in various systems, such as ferromagnetic UGe_2 ¹¹, liquid crystals¹², and $Pb_{1-x}Ge_xTe$ alloys¹³. This strongly suggests that the critical phenomena and its physical models are universal. In addition, anomalous reduction in thermal conductivity κ has been observed during critical phase transitions for a lot of materials, such as Cu_2Se ^{14, 15} and $BaTiO_3$ ¹⁶. Dramatic increment in electrical resistivity has been reported around the ferroelectric phase transitions of $SnTe$ ^{7, 8}, $PbTe$ ¹⁷, and $GeTe$ ⁶. We have added a few sentences into the revised manuscript.

Comment 8: *In Eq. (2), why do the authors used the classical Landau theory for the distribution function of the band energy E , instead of the quantum theory?*

Reply: The Landau theory describes the relationship between the order parameter ξ and the energy $\Delta\Phi$. Herein we assumed that the variation of order parameters causes perturbations in the energy band, and phenomenologically applied Taylor series expansion to the band-edge energy E near ξ_0 . Then we established a connection between order parameter ξ and band-edge energy E (Eq. 2).

We can also calculate the energy band for different structures based on the quantum theory, just like the energy band structures shown in Figure 2. However, these calculation processes are very complex and time-consuming, and it is challenging to establish the connection between ξ and E . Therefore, to simply the model, here we use the classical Landau theory.

References

1. D. R. Brown, T. Day, K. A. Borup, S. Christensen, B. B. Iversen and G. J. Snyder, *APL Mater.*, 2013, **1**, 052107.
2. H. Liu, X. Yuan, P. Lu, X. Shi, F. Xu, Y. He, Y. Tang, S. Bai, W. Zhang, L. Chen, Y. Lin, L. Shi, H. Lin, X. Gao, X. Zhang, H. Chi and C. Uher, *Adv. Mater.*, 2013, **25**, 6607-6612.
3. K. Zhao, A. B. Blichfeld, H. Chen, Q. Song, T. Zhang, C. Zhu, D. Ren, R. Hanus, P. Qiu, B. B. Iversen, F. Xu, G. J. Snyder, X. Shi and L. Chen, *Chem. Mater.*, 2017, **29**, 6367-6377.
4. E. Eikeland, A. B. Blichfeld, K. A. Borup, K. Zhao, J. Overgaard, X. Shi, L. Chen and B. B. Iversen, *IUCrJ*, 2017, **4**.
5. R. S. Bauer and B. A. Huberman, *Phys Rev B*, 1976, **13**, 3344.
6. I. Avramova and S. Plachkova, *J. Phys. Condens. Matter* 2001, **13**, 43.
7. K. Kobayashi, Y. Kato, Y. Katayama and K. Komatsubara, *Solid State Commun.*, 1975, **17**, 875-878.
8. G. Minemura and A. Morita, *Solid State Commun.*, 1978, **28**, 273-275.
9. R. Chen, P. Qiu, B. Jiang, P. Hu, Y. Zhang, J. Yang, D. Ren, X. Shi and L. Chen, *J. Mater. Chem. A*, 2018, **6**, 6493-6502.
10. K. Zhao, K. Liu, Z. Yue, Y. Wang, Q. Song, J. Li, M. Guan, Q. Xu, P. Qiu, H. Zhu, L. Chen and X. Shi, *Adv Mater*, 2019, **31**, e1903480.
11. N. Tateiwa, T. Kobayashi, K. Amaya, Y. Haga, R. Settai and Y. Ōnuki, *Phys. Rev. B*, 2004, **69**, 180513.
12. H. Haga, Z. Kutnjak, G. S. Iannacchione, S. Qian, D. Finotello and C. W. Garland, *Phys. Rev. E*, 1997, **56**, 1808-1818.
13. N. Sugimoto, T. Matsuda and I. Hatta, *J. Phys. Soc. Jpn.*, 1981, **50**, 1555-1559.
14. H. Liu, X. Yuan, P. Lu, X. Shi, F. Xu, Y. He, Y. Tang, S. Bai, W. Zhang and L. Chen, *Adv. Mater.*, 2013, **25**, 6607-6612.
15. H. Liu, X. Shi, M. Kirkham, H. Wang, Q. Li, C. Uher, W. Zhang and L. Chen, *Mater. Lett.*, 2013, **93**, 121-124.
16. A. Mante and J. Volger, *Phys. Lett. A*, 1967, **24**, 139-140.
17. R. M. Murphy, É. D. Murray, S. Fahy and I. Savić, *Phys Rev B*, 2017, **95**, 144302.

Summary of the changes

1. One sentence is added on page 10 and one figure (i.e. Fig. S1a) is added on page 7 of the Supplementary Information to demonstrate the reversible phase transition of Cu₂Se.
2. One paragraph is added on page 10 to discuss the biases or oversights of the model.
3. One sentence is added in 2nd paragraph of page 11 to discuss the power factor.
4. Two sentences are added on page 12 to discuss the reduction of phase transition temperature.
5. Two sentences are added in the 2nd paragraph of page 13 to explain the reduced hole concentration in S alloyed Cu₂Se.
6. One sentence is added in 1st paragraph of page 15 to discuss the lattice thermal conductivity.
7. Fig. 4 is restructured and the corresponding caption is modified on page 16.
8. Six sentences are added on pages 17 and 18 to discuss the transferability of our findings across different areas.
9. Four sentences are added in the 2nd paragraphs on page 18 to prospect the strategies for tuning and broadening phase transition temperature.
10. The electrical measurement details are added in the Methods section.
11. Three sentences are added in the Methods section and one figure (i.e. Fig. S6) is add in the Supplementary Information to discuss the variation of unit cell volume.

12. Two figures including the power factor and lattice thermal conductivity are added in the Fig. S2 of the Supplementary Information.
13. The schematic of the home-made equipment is added in Fig. S5.

We sincerely thank the three reviewers again for their careful consideration and very helpful suggestions as well as comments. We believe that we have made clear explanations and proper modifications.

Sincerely yours,

Xun Shi

Professor Xun Shi

Shanghai Institute of Ceramics, Chinese Academy of Sciences,
and School of Material Science and Engineering, Shanghai Jiao Tong University

Tel: +86-138-1871-0650

E-mail: xshi@mail.sic.ac.cn (or xshi@sjtu.edu.cn)

REVIEWERS' COMMENTS

Reviewer #1 (Remarks to the Author):

The authors had well responded the questions and revised the manuscript. I think it can be accepted by Nature Communications.

Reviewer #3 (Remarks to the Author):

I recommend the manuscript for publication in its current form.

Referee#1

Comments to the Author: *The authors had well responded the questions and revised the manuscript. I think it can be accepted by Nature Communications.*

Reply: We sincerely thank the reviewer again for the careful consideration and very helpful suggestions as well as comments.

Referee#3

Comments to the Author: *I recommend the manuscript for publication in its current form.*

Reply: We sincerely thank the reviewer again for the careful consideration and very helpful suggestions as well as comments.

Sincerely yours,

Xun Shi

Professor Xun Shi

Shanghai Institute of Ceramics, Chinese Academy of Sciences,

and School of Material Science and Engineering, Shanghai Jiao Tong University

Tel: +86-138-1871-0650

E-mail: xshi@mail.sic.ac.cn (or xshi@sjtu.edu.cn)